# Viral dark matter and virus–host interactions resolved from publicly available microbial genomes

**Simon Roux[1][†], Steven J Hallam[2,3], Tanja Woyke[4], Matthew B Sullivan[1]\*[†][‡]**

[1]Department of Ecology and Evolutionary Biology, University of Arizona, Tucson, United States; [2]Department of Microbiology and Immunology, University of British Columbia, Vancouver, Canada; [3]Graduate Program in Bioinformatics, University of British Columbia, Vancouver, Canada; [4]U.S Department of Energy Joint Genome Institute, Walnut Creek, United States

**\*For correspondence:** mbsulli@ gmail.com

**Present address:** [†]Department of Microbiology, The Ohio State University, Columbus, United States; [‡]Department of Civil, Environmental, and Geodetic Engineering, Columbus, United States

**Competing interests:** The authors declare that no competing interests exist.

**Abstract** The ecological importance of viruses is now widely recognized, yet our limited knowledge of viral sequence space and virus–host interactions precludes accurate prediction of their roles and impacts. In this study, we mined publicly available bacterial and archaeal genomic data sets to identify 12,498 high-confidence viral genomes linked to their microbial hosts. These data augment public data sets 10-fold, provide first viral sequences for 13 new bacterial phyla including ecologically abundant phyla, and help taxonomically identify 7–38% of 'unknown' sequence space in viromes. Genome- and network-based classification was largely consistent with accepted viral taxonomy and suggested that (i) 264 new viral genera were identified (doubling known genera) and (ii) cross-taxon genomic recombination is limited. Further analyses provided empirical data on extrachromosomal prophages and coinfection prevalences, as well as evaluation of in silico virus–host linkage predictions. Together these findings illustrate the value of mining viral signal from microbial genomes.

## Introduction

Over the past two decades, our collective understanding of microbial diversity has been profoundly expanded by cultivation-independent molecular methods (*Pace, 1997*; *Whitman et al., 1998*; *Rappé and Giovannoni, 2003*; *DeLong, 2009*; *Hanson et al., 2012*). It is now widely recognized that interconnected microbial communities drive matter and energy transformations in natural and engineered ecosystems (*Falkowski et al., 2008*), while also contributing to health and disease states in multicellular hosts (*Clemente et al., 2012*). Concomitant with this changing worldview is a growing awareness that viruses modulate microbial interaction networks and long-term evolution with resulting feedbacks on ecosystem functions and services (*Suttle, 2007*; *Rodriguez-Valera et al., 2009*; *Forterre and Prangishvili, 2013*; *Hurwitz et al., 2013*; *Brum et al., 2014*; *Brum and Sullivan, 2015*).

However, our understanding of viral diversity and virus–host interactions remains a major bottleneck in the development of predictive ecosystem models and unifying eco-evolutionary theories. This is because the lack of a universal marker gene for viruses hinders environmental survey capabilities, while the number of isolate viral genomes in databases remains limited: for comparison, more than 25,000 bacterial and archaeal host genomes are available in NCBI RefSeq (January 2015), whereas only 1,531 of their viruses were entirely sequenced and most (86%) of these derive from only 3 of 61 known host phyla (*Roux et al., 2015a*). Thus, although advances in high-throughput sequencing expand the bounds of viral sequence space, these data sets are dominated by uncharacterized sequences (usually 60–95%), termed 'viral dark matter' (*Reyes et al., 2012*;

**eLife digest** Viruses are infectious particles that can only multiply inside the cells of microbes and other organisms. Little is known about the genetic differences between virus particles (so-called 'genetic diversity'), especially compared to what we know about the diversity of bacteria, archaea, and other single-celled microbes. This lack of knowledge hampers our understanding of the role viruses play in the evolution of microbial communities and their associated ecosystems.

Studying the genetics of the viruses in these communities is challenging. There is no single 'marker' gene that can be used to identify all viruses in environmental samples. Also, many of the fragments of viral genomes that have been identified have not yet been linked to their host microbes. Many viruses integrate their genome into the DNA of their host cell, and there are computational tools available that exploit this ability to identify viruses and link them to their host. However, other viruses can live and multiply inside cells without integrating their genome into the host's DNA.

Earlier in 2015, researchers developed a new computational tool called VirSorter that can predict virus genome sequences within the DNA extracted from microbes. VirSorter identifies viral genome sequences based on the presence of 'hallmark' genes that encode for components found in many virus particles, together with a reference database of genomes from many viruses.

Now, Roux et al.—including some of the researchers from the earlier work—use VirSorter to predict viral DNA from publicly available bacteria and archaea genome data. The study identifies over 12,000 viral genomes and links them to their microbial hosts. These data increase the number of viral genome sequences that are publically available by a factor of ten and identify the first viruses associated with 13 new types of bacteria, which include species that are abundant in particular environments.

It is possible for several different viruses to infect a single cell at the same time. Some viruses are known to be able to exchange DNA, and if this happens frequently in other viruses, it could have a big impact on how viruses evolve. Roux et al.'s findings suggest that although it is common for several different viruses to infect the same cell, it is relatively rare for these viruses to exchange genetic material.

Roux et al.'s findings demonstrate the value of searching publicly available microbial genome data for fragments of viral genomes. These new viral genomes will serve as a useful resource for researchers as they explore the communities of viruses and microbes in natural environments, the human body and in industrial processes.

*Youle et al., 2012*; *Mizuno et al., 2013*; *Brum and Sullivan, 2015*). In the absence of closely related isolates, viral genes and genomes remain unlinked to hosts, which greatly limits ecological and evolutionary inferences.

Alternatively, viral sequence space can be explored in a known host context by revealing putative viral sequences hidden in microbial genomes. Such signal was first analyzed through annotation of prophages—viral genomes integrated in microbial genomes. Numerous tools exist to automatically detect prophages (*Fouts, 2006*; *Lima-Mendez et al., 2008a*; *Zhou et al., 2011*; *Akhter et al., 2012*), so prophage diversity and abundance are relatively well studied (*Casjens, 2003*; *Canchaya et al., 2004*). Early estimations, when only a few hundred bacterial genomes were available, suggested that prophages are common (62% of bacterial genomes tested contained at least one), existing as intact and functional forms or in varying degrees of decay (*Casjens, 2003*). Given that tens of thousands more microbial genomes are now publicly available, it is expected that many new prophages and other viral sequences remain to be discovered.

Further, other viral signals might be prevalent in modern microbial genomic data sets. First, certain types of prophage do not integrate into the host genome. These 'extrachromosomal prophages' (also termed 'plasmid prophage') exist outside the microbial chromosome until induced to undergo lytic replication. These have been known to occur for decades (e.g., coliphage P1, *Sternberg and Austin, 1981*), though their abundance in nature is unknown. Second, some phages can enter a 'chronic' cycle, in which they replicate in the cell outside of the host chromosome, and produce virions that are extruded without killing their host (*Abedon, 2009*; *Rakonjac et al., 2011*). Third, a phage

'carrier state' has been observed, in which a lytic phage is maintained and multiplied within a cultivated host population without measurable effect on cell growth (*Bastías et al., 2010*). This phenomenon is thought to arise due to the presence of both resistant and sensitive cells that frequently transition between these two states. Sometimes also termed 'partial resistance', such states that enable the coexistence of phage and host in culture have now been observed in different systems (*Vibrio*, *Escherichia coli*, *Salmonella*, *Flavobacterium*), and are linked to slight decreases in growth rate or cell concentration but no host cell clearing as would be observed for 'typical' lytic viruses (i.e., plaque formation), thus could go unnoticed in a microbial cell culture (*Fischer et al., 2004*; *Carey-Smith et al., 2006*; *Middelboe et al., 2009*). All three of these lesser studied types of infection would result in the assembly of viral sequences outside of the main host chromosome in a microbial genome sequencing project and could be a new type of viral signal in modern microbial genomic data sets due to deep sequencing and public release of draft (i.e., not completely assembled) genomic sequences.

Finally, single amplified genome (SAG) data sets, sourced from anonymously sorted, amplified, and sequenced cells, are especially valuable for accessing the vast majority of environmental microbes that remain uncultivated in the lab (*Rinke et al., 2013*; *Kashtan et al., 2014*). Single-cell amplified genomes can reveal viral sequences directly linked to uncultivated hosts (*Yoon et al., 2011*; *Roux et al., 2014*; *Labonté et al., 2015*). When combined with metagenomic sequences, these data provide information on population dynamics, lineage-specific viral-induced mortality rates, relative ratios of prophages and current lytic infections, as well as putative links between viral infection and host metabolic state (*Roux et al., 2014*; *Labonté et al., 2015*). Thus, as microbial genomic data sets evolve from complete genomes to fragmented draft and single-cell genomes, new windows into viral diversity and virus–host interactions are opened.

Here, we applied a recently developed and automated virus discovery pipeline, VirSorter (*Roux et al., 2015a*), to mine the viral signal from 14,977 publicly available bacterial and archaeal genomic data sets. This identified 12,498 high-confidence viral sequences with known hosts, ~10-fold more than in the RefSeqVirus database, that we then used to expand our understanding of viral diversity and virus–host interactions.

## Results and discussion

### New viruses detected in public microbial genomic data sets with VirSorter

VirSorter is designed to predict bacterial and archaeal virus sequences in isolate or single-cell draft genomes, as well as complete genomes (*Roux et al., 2015a*). Briefly, VirSorter identifies viral sequences through (i) statistical enrichment in viral gene content, using a reference database composed of viral genomes of archaeal and bacterial viruses from RefSeq (hereafter named RefSeqABVir for 'RefSeq Archaea and Bacteria Viruses') and assembled from viral metagenomes (database 'Viromes' in VirSorter), or (ii) a combination of viral 'hallmark' gene(s) that code for virion-related functions such as major capsid proteins or terminases (*Koonin et al., 2006*; *Roux et al., 2014*), and at least one viral-like genomic feature: statistical depletion in genes with a hit in the PFAM database, statistical enrichment in uncharacterized genes, short genes, or strand bias (i.e., consecutive genes which tend to be coded on the same strand).

Applied to 14,977 publicly available microbial genomes (*Figure 1—source data 1*), VirSorter identified 12,498 high-confidence viral sequences representing either long genome fragments (>10 kb when linear) or complete genomes (contigs detected as circular). These viral sequences were found in 5492 of the microbial genomes (~30%). Simply scanning the identified viruses for novel hosts extended the host range of common viral families to now include several recently described phyla like *Caldiserica* (formerly known as candidate phylum OP5), *Marinimicrobia* (SAR406 also known as Marine Group A), or *Omnitrophica* (OP3), in addition to other understudied groups such as *Poribacteria*, *Nitrospinae*, *Cloacimonetes* (WWE1), and Chloroflexi-type SAR202 (*Figure 1*, *Figure 1—source data 2*, *Figure 1—source data 3*). Uncovering the first viruses infecting these major microbial groups is critical given that many candidate phyla are abundant in understudied ecosystems and play substantial roles in coupled biogeochemical cycling (*Wright et al., 2012*; *Wrighton et al., 2012*; *Castelle et al., 2013*; *Kamke et al., 2013*; *Rinke et al., 2013*; *Allers et al., 2013b*; *Emerson et al., 2015*).

BLAST-based family-level affiliations suggested that 90.45% of these 12,498 sequences correspond to *Caudovirales*, 6.82% to ssDNA viruses (predominantly *Inoviridae* family), and 2.73% could not be

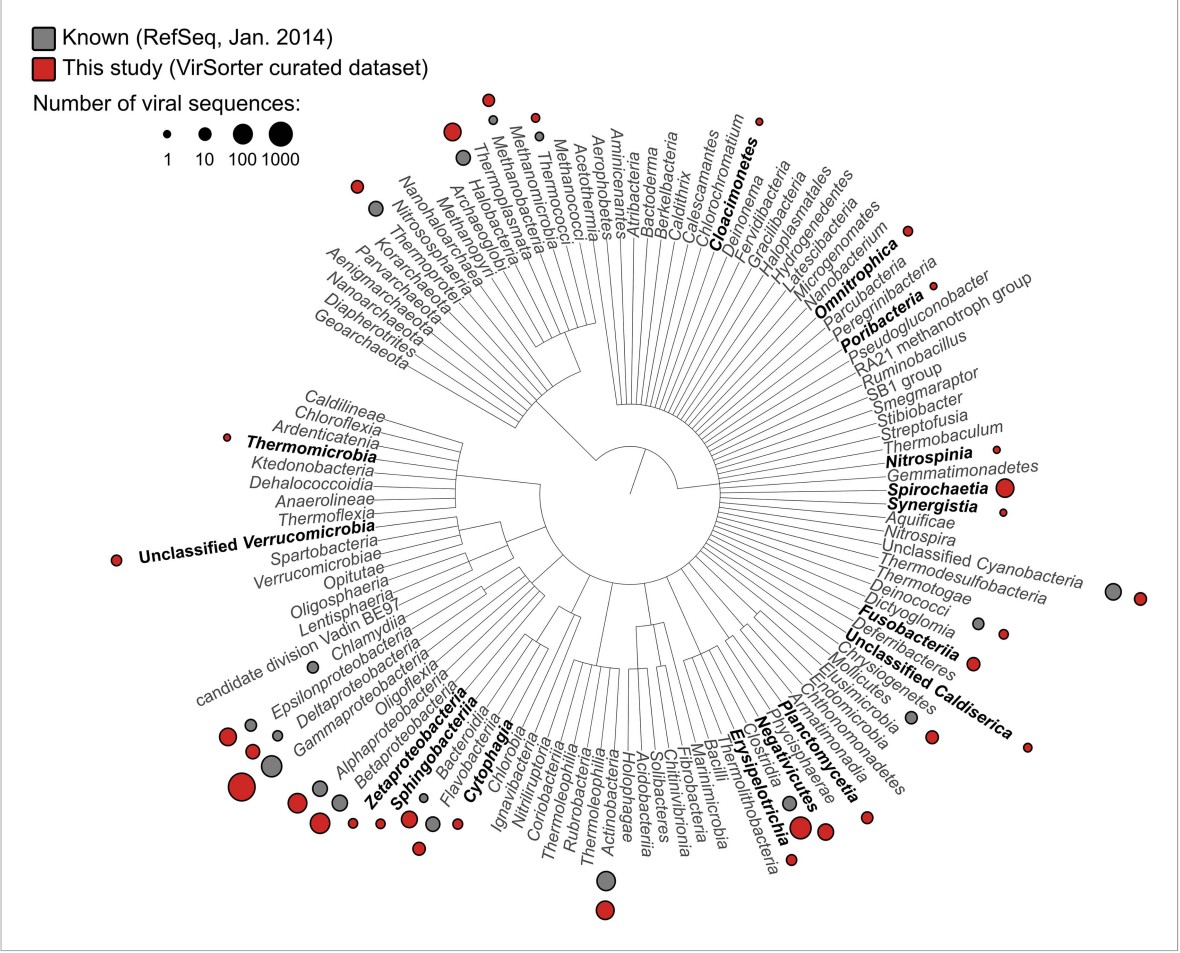

**Figure 1**. Distribution of viral sequences from the VirSorter curated data set across the bacterial and archaeal phylogeny. For each bacteria or archaea phylum (or phylum-level group), corresponding viruses in RefSeq (gray) and VirSorter curated data set (red) are indicated with circles proportional to the number of sequences available. Groups for which no viruses were available in RefSeq are highlighted in black.

The following source data and figure supplements are available for figure 1:

**Source data 1**. List of data sets mined for viral signal.

**Source data 2**. New virus–host associations detected in VirSorter sequences.

**Source data 3**. Summary table of VirSorter data set sequences.

**Figure supplement 1**. Viral diversity in the VirSorter data set.

**Figure supplement 2**. Genome map comparison (**A**) and recruitment plot (**B**) of *Bacteroidia* virus sequences from a putative new order.

confidently assigned (*Figure 1—figure supplement 1*). Among the unassigned group, 7 sequences lacked any hit to a viral reference genome. These 7 short (4.1 kb) near-identical circular contigs from *Bacteroides* draft genomes were detected as viral based on sequence similarity with human gut viromes, but contained two genes associated with plasmid replication (*Figure 1—figure supplement 2A*). This could suggest a plasmid origin, but the high and even coverage of these genomes across several CsCl-purified viromes from different studies (*Kim et al., 2011*; *Minot et al., 2012*) suggests that they are derived from encapsidated particles typical of viruses (*Figure 1—figure supplement 2B*). If confirmed, these sequences would represent the first complete genomes for an entirely new viral order.

## 264 new putative viral genera identified through genome-based network clustering

To better determine relationships between viral genomes and host range, we next built a network based on shared gene content to quantify genetic relatedness between the 12,498 sequences identified with VirSorter and the 1,240 taxonomically curated genomes available in RefSeqABVir (*Figure 2—figure supplement 1* and see 'Materials and methods'). Despite the absence of a universal marker gene, a long history of organizing viral sequence space through genome-to-genome comparison exists using either gene content (*Rohwer and Edwards, 2002*; *Lima-Mendez et al., 2008b*) or nucleotide composition (*Sims and Jun, 2009*; *Labonté and Suttle, 2013*). We used MCL (Markov Cluster Algorithm) based on the number of shared genes between sequence pairs as it had been previously shown to accurately recapitulate taxonomic relationships in the *Caudovirales*, which dominated our data set (*Enright et al., 2002*; *Lima-Mendez et al., 2008b*).

Most (99.3% of 12,498) sequences affiliated to one of 614 virus clusters (VCs), of which 535 contained at least one complete genome or large genomic fragment (>30 kb), and approximately half (271 of 535 VCs) included RefSeqABVir sequences (*Figure 2A*, *Figure 2—source data 1*). Those VCs with RefSeqABVir sequences provided the opportunity to evaluate whether a VC corresponded to any particular taxonomic level of ICTV classification. Of 43 RefSeq-curated viral genera, 27 have all their sequences in the same VC, 12 were spread across two VCs, and 4 were spread across >2 VCs—these latter genera included the Spouna-like viruses (3 VCs), N4-like viruses (4 VCs), Lambda-like viruses (9 VCs), and Inoviruses (11 VCs). Consistent with previous applications of this method, VCs identified in this analysis were thus approximately equivalent to a RefSeq-curated viral genus (*Lima-Mendez et al., 2008b*).

Given this level of taxonomic resolution and ignoring the 79 VCs that lacked large (>30 kb) genome sequences, we identified a total of 264 new candidate viral genera (i.e., 264 VCs with no sequences from RefSeqABVir, *Figure 2—source data 1*). These 264 candidate genera were derived from both understudied and well-studied hosts (e.g., *Gammaproteobacteria* and *Bacilli*, *Figure 2B*) and included 5 of the 30 highest-membership VCs (*Figure 2—source data 1*), which confirms that our knowledge of viral diversity is limited even in well-studied hosts and with prevalent viruses.

## VirSorter curated data set includes extrachromosomal genomes and improves virome affiliation

Of the 12,498 sequences, 5,232 were prophages (i.e., a viral genome integrated into a microbial contig) and 1,756 were either complete (circularized) or large (>30 kb) genome fragments assembled outside of the host chromosome (i.e., no microbial gene was detected on the contig, *Figure 3A*, *Figure 1—source data 1*).

To estimate how often a large (>30 kb) genome fragment could be an integrated prophage and not capture the microbial gene content, we simulated the process for 22 different prophage-containing bacterial genomes 'sequenced' (in silico) at coverage of 5, 25, 50, 75, and 100× (see 'Material and methods'). These analyses suggested that only 2.3% of large (>30 kb) prophage-originating contigs lacked any identifiable microbial genes. Thus, these latter 1,756 sequences must largely be extrachromosomal sequences and so represent a unique data source for quantifying the prevalence of under-studied viral infection modes including chronic infections, lytic viruses, or extrachromosomal prophages.

Although we identified no clear sequence-based marker for the first two infection types, we could conservatively estimate the fraction of extrachromosomal prophages by identifying plasmid partition genes (ParA and ParB, *Davis et al., 1992*; *Saint Girons et al., 2000*; see *Figure 3—figure supplement 1* for an example of a putative extrachromosomal prophage displaying ParA-ParB genes). These genes were significantly more abundant in the 1,756 circular and large genome fragments than in the rest of the data set (13% vs 1%, respectively; poisson test p-value $< 10^{-05}$). Thus, at least 13% of these sequences appear to be bona fide extrachromosomal prophages, whereas the others might be lytic viruses in 'carrier' states, chronic infections, or extrachromosomal prophages without detectable ParA/ParB genes.

Beyond this glimpse into under-studied viral infection modes, these new reference genomes are likely to help improve taxonomic affiliation for the 'viral dark matter' in viromes. To quantify this, we added these sequences to the RefSeqABVir database and assigned taxonomy to predicted genes in

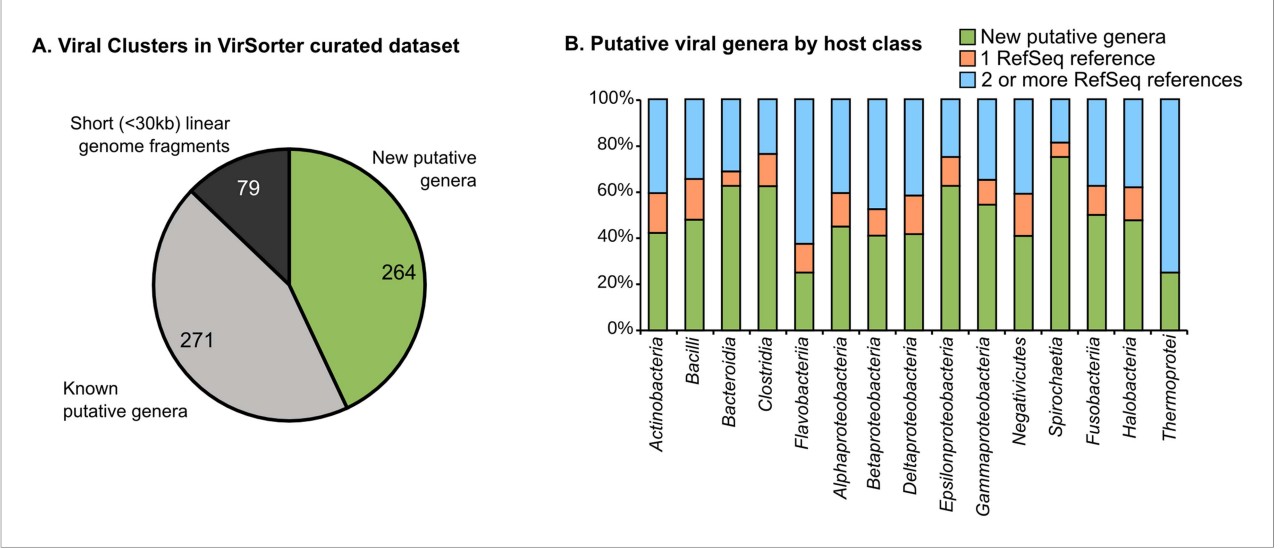

**Figure 2**. Degree of novelty of viruses detected in VirSorter curated data set. (**A**) Viral clusters (VCs) are considered as putative new genera when including at least one sequence larger than 30 kb, circular, or known to be a complete genome (from RefSeq). These putative genera were considered as 'new' when the VC did not include any RefSeq sequence, and 'known' otherwise. (**B**) The proportion of new VCs (containing no RefSeqABVir), VCs with only one RefSeqABVir sequence, and VCs with more than one RefSeqABVir sequence is displayed for host classes associated with more than 10 virl sequences. Only 'putative genera' VCs were considered (i.e., clusters containing a RefSeqABVir genome, a circular sequence, or a sequence with more than 30 predicted genes).

The following source data and figure supplements are available for figure 2:

**Source data 1**. Summary table of virus clusters (VCs).

**Figure supplement 1**. Structure of viral sequence space sampled in VirSorter data set.

**Figure supplement 2**. Benchmarks used to determine the best value for inflation and significance thresholds for virus clustering.

three large-scale virome data sets available. We found that the VirSorter curated data set improved affiliation by 32 and 40%, respectively, in the marine Pacific Ocean Viromes (POV) (*Hurwitz and Sullivan, 2013*) and Tara Oceans Viromes (TOV) (*Brum et al., 2015*) data sets, and more than doubled the number of affiliated genes in human gut viromes (*Minot et al., 2012*, *Figure 3B*). This particularly strong improvement in the human gut virome affiliation is presumably due to enterobacteria being abundant among current publicly available microbial genomes.

Finally, both the detection of non-integrated viral genomes and the improved virome affiliation suggest that the VirSorter curated data set includes not only integrated prophage data, but also viruses actively infecting these microbes (i.e., not integrated in the host chromosome and producing virions) with under-studied infection modes.

## Long-term evolutionary patterns of bacterial and archaeal virus genomes

Examination of the VCs network beyond classification revealed additional higher order patterns. First, bacterial and archaeal viruses clustered separately in >99% of VCs; the exception (VC_89) included a single and unique (*Garrett et al., 2010*) archaeal virus (Hyperthermophilic Archaeal Virus 2, NC_014321) that clustered with 21 bacterial viruses, presumably due to poor archaeal virus representation. Second, >95% of these VCs contained exclusively one nucleic acid type (e.g., DNA or RNA, and dsDNA or ssDNA, *Figure 2—figure supplement 1*), although RNA viral representation is low because only RefSeq-curated families *Cystoviridae* and *Leviviridae* were available (no RNA viruses were detected with VirSorter, presumably because DNA-based data sets were analyzed). The 15 VCs including both ssDNA and dsDNA viral genomes are either associated with archaeal viruses for which composite ssDNA/dsDNA genomes were already described (2 VCs; *Sencilo et al., 2012*) or more

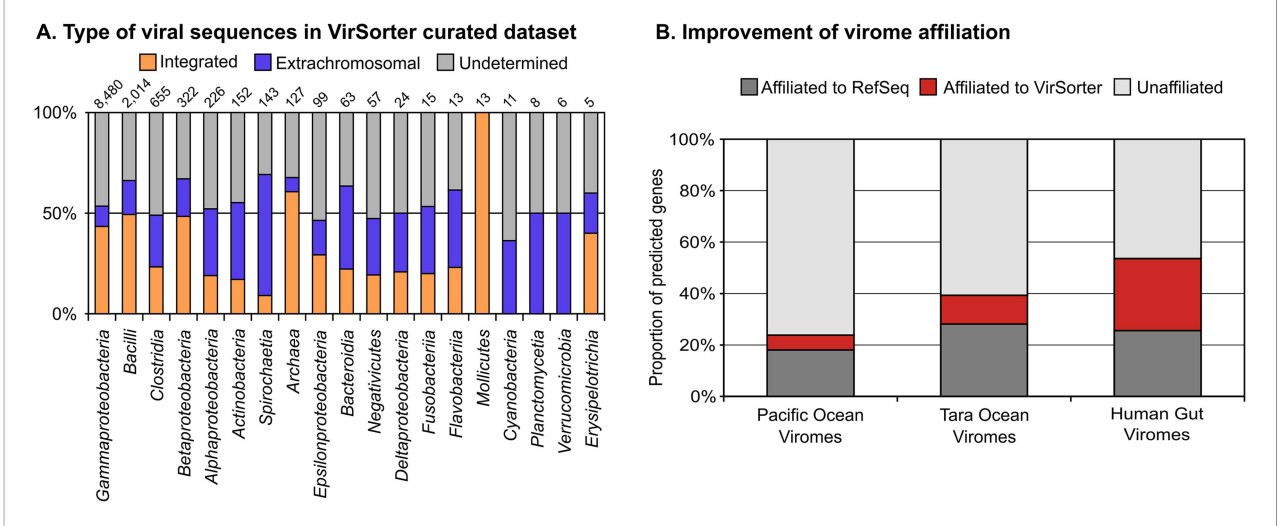

**Figure 3**. Extrachromosomal prophages in VirSorter curated data set and improvement in virome affiliation. (**A**) The distribution of VirSorter curated data set as 'integrated' (i.e., prophages integrated in the host chromosome), 'extrachromosomal' (i.e., >30 kb or circular sequences with no microbial genes), or 'undetermined' (<30 kb linear with no microbial genes) is indicated for each host class with at least five VirSorter curated data set sequences. The number of sequences associated with each host class in indicated above the histogram. (**B**) Improvement in the proportion of affiliated genes from viromes with VirSorter data set. Predicted genes from the Pacific Ocean Viromes (*Hurwitz and Sullivan, 2013*), Tara Ocean Viromes (*Brum et al., 2015*), and Human Gut Viromes (*Minot et al., 2012*) were compared to RefSeqVirus (May 2015) and the VirSorter data set (BLASTp, threshold of 50 on bit score and 0.001 on e-value). Predicted proteins affiliated to VirSorter (in blue) did not display any significant similarity to a RefSeq sequence.

The following figure supplement is available for figure 3:

**Figure supplement 1**. Contig map of a putative new extrachromosomal prophage.

surprisingly with ssDNA *Inoviridae*, which clustered with *Caudovirales* in 13 VCs (*Figure 2—source data 1*). For 9 of these 13 *Inoviridae*–*Caudovirales* VCs, some of the sequences were wrongly affiliated due to genes shared by both viral families such as integrases, exonucleases, and replication-associated proteins. Two other VCs corresponded to prophage sequences that include genes similar to *Inoviridae* and *Caudovirales* and could actually be two different viruses integrated at the same genome location. However, the 2 remaining mixed VCs (VC_128 and VC_215) include sequences displaying a mix of *Caudovirales* and *Inoviridae* genes (VC_215 sequences also included *Corticoviridae* genes). We posit that these might represent new composite genomes beyond the ones already described for archaea viruses (*Sencilo et al., 2012*) and the recently discovered RNA–DNA chimeric viruses (*Diemer and Stedman, 2012*; *Roux et al., 2013*; *Krupovic et al., 2015*).

We next evaluated the scale and range of viral co-infection, a phenomenon critical to viral genome evolution and thought to blur this vertical gene inheritance signal used to classify genomes into VCs. Indeed, the fact that super-infection of prophage-containing bacteria would provide genomic proximity for gene acquisition via illegitimate recombination was posited more than a decade ago (*Mosig, 1998*). However, viral co-infection rates remain unconstrained with the only data for natural systems derived from a single large-scale single-cell genomic data set where ~35% of infected cells contained multiple viruses (*Roux et al., 2014*). Here, in the 5492 microbial genomes with detectable viral signal, nearly half (2445) contained more than one detectable virus (*Figure 4*). Most (~82%) of these co-infections involved multiple *Caudovirales*, as previously observed (*Casjens, 2003*), and likely provides mechanism for viral gene exchange and may be more common in some phages displaying rampant mosaicism (e.g., the *Siphoviridae*, *Hendrix et al., 1999*) than others. The second most commonly observed co-infections (9%) occurred between ssDNA *Inoviridae* and dsDNA *Caudovirales* (*Figure 4*). These genomes represented the mixed VCs from the network analyses and putative new composite genomes described above. Mechanistically, *Inoviridae* might be more prone to such co-infection due to their long infection cycle whereby they extrude their

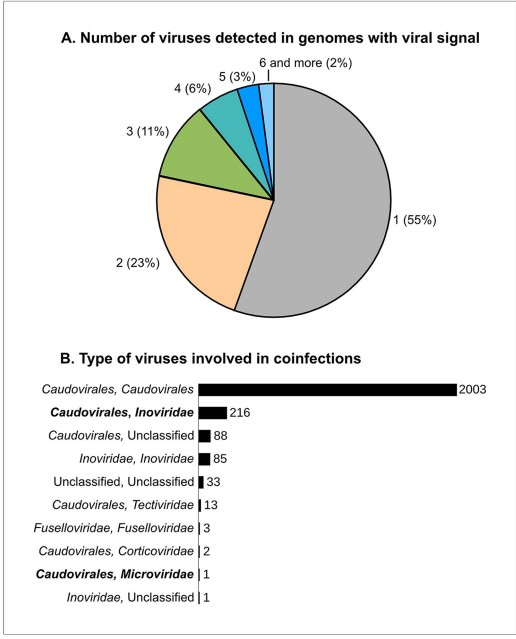

**Figure 4**. Scale and range of co-infection. (**A**) Number of different viral sequences detected by host genome. Numbers are based on the set of microbial genomes with at least one viral sequence detected (5492 genomes). (**B**) Affiliation of viruses involved in multiple infections of the same host. Affiliations are deduced from best BLAST hits alongside the viral sequences, as in *Figure 1*. Co-infections involving dsDNA and ssDNA viruses are highlighted in bold.

filamentous virions without killing their host (*Rakonjac et al., 2011*), with a dsDNA replication stage (*Salim et al., 2008*) that could increase genomic exchanges with co-infecting dsDNA viruses.

Together, these findings suggest that genome-based network analyses could be used to identify novel viruses, as well as to infer host domain (archaeal or bacterial, >99% accuracy) and nucleic acid type (ssDNA or dsDNA, >95% accuracy). Evolutionarily, we posit that while co-infection by multiple viruses appears common, the consistency of so many VCs with ICTV taxonomy suggests that most phage genomes harbor a largely vertically inherited core gene set as detected for marine T4-like populations (*Ignacio-Espinoza and Sullivan, 2012*; *Marston et al., 2012*; *Deng et al., 2014*) rather than the rampant mosaicism paradigm largely derived from *Siphoviridae* genomes (*Hendrix et al., 1999*). While data remain limited to a subset of the known microbial phyla, it might be that viral infection modes influence the tempo of their genome evolution. Specifically, we posit that horizontal gene transfer is more prevalent in phages that occupy host cells longer due to lysogenic or chronic infection stages and/or infect densely packed hosts (e.g., biofilms or clumped life stages) as these parameters would increase the probability of co-infection. Perhaps then, at least for more highly lytic viral groups, genome-based clustering approaches can now be leveraged for their taxonomic predictive value as suggested over a decade ago (*Rohwer and Edwards, 2002*).

## Global virus–host network is confirmed as modular

Beyond charting diversity and taxonomic affiliation of viral sequence space, the VirSorter data set provided a unique opportunity to explore virus–host interactions. Beyond the above-noted expansion of viruses to novel hosts, we next examined these patterns on a global scale by constructing a virus–host interaction network based on database-available taxa. When considering viral diversity at the genus level, the network displays a modular topology (*Figure 5* and *Figure 5—figure supplement 1*). Such modularity in virus–host interaction networks suggests that hosts are specifically associated with particular viruses (*Weitz et al., 2012*), probably reflecting long-term coevolution between microbial hosts and their viruses. Such modular structure was expected, but not observed in previous virus–host interaction network studies, likely due to the short phylogenetic distances between hosts evaluated in available data sets (*Flores et al., 2011*). The modular network presented here derives from a data set spanning 18 phyla across bacterial and archaeal domains. These results confirmed the prediction that 'at macroevolutionary scales, host–phage interaction matrices should be typified by a modular structure' (*Flores et al., 2011*), as also had been observed across 215 phage types against 286 host types of unknown diversity (*Flores et al., 2013*).

## Virus–host adaptation signals detected at the genome composition and codon usage level

Finally, given the number of virus–host linkages revealed by VirSorter, we evaluated the adaptation of viral genome composition within the host milieu—an idea practiced in the literature with limited

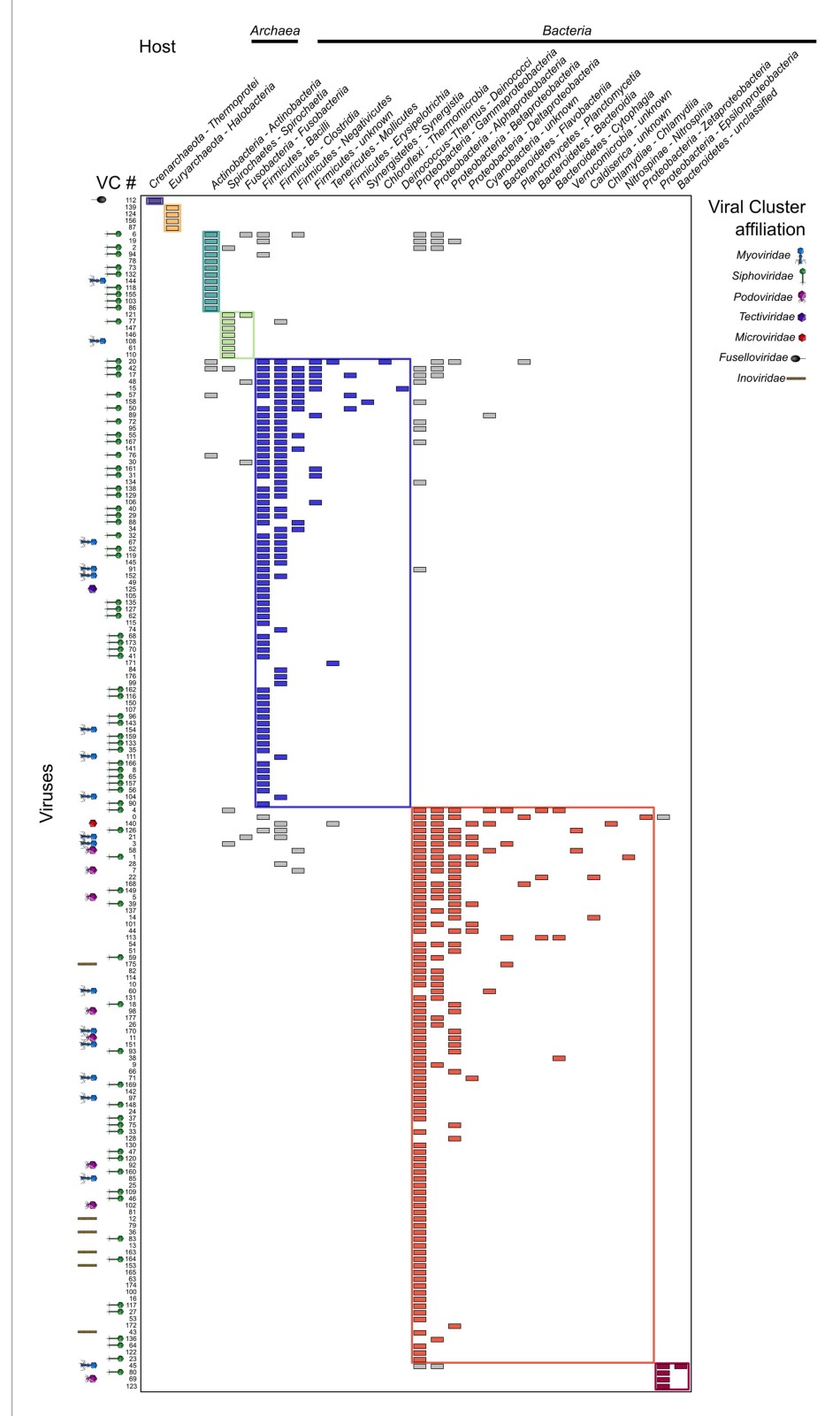

**Figure 5**. Virus–host network between virus clusters and host classes (matrix visualization). A cell in the matrix is colored when at least one virus from a virus cluster (VC, rows) was retrieved in a genome from a host class (columns). This virus–host network is detected as significantly modular by lp-Brim (modularity Q = 0.45; the same index computed from 99 randomly permuted matrices ranged from 0.02 to 0.17, with an average of 0.08). The different

*Figure 5. Continued*

modules are highlighted in color, with inter-module links in gray. Virus clusters are identified by their number and their family-level affiliation (based on BLAST-based affiliation of the cluster members) is indicated next to each cluster when available (virus clusters with inconsistent members affiliation are considered as 'unclassified', affiliations are spread along the x-axis for spacing purpose). Host phylum and class are indicated for each host column, with domains indicated above the corresponding hosts.

The following figure supplement is available for figure 5:

**Figure supplement 1**. Virus–host network between virus clusters and host classes (network visualization).

genomic information (*Pride et al., 2006*; *Carbone, 2008*; *Cardinale and Duffy, 2011*). To this end, we computed the distance between viral and microbial genomes in terms of mono-, di-, tri-, tetra-nucleotide frequency and codon usage, and compared the distances between the virus and its host vs non-hosts in the data set. Every metric tested displayed a smaller distance between viruses and their hosts than with non-host genomes, with tetranucleotide frequency (TNF) maximizing the host to non-host distances (*Figure 6*).

Among dsDNA viruses, host-correlated genome composition patterns were robust across integrated prophages and extrachromosomal genomes (i.e., viral sequences assembled outside of the main host chromosome, *Figure 6A*, *Figure6—figure supplement 1*). Our expectations were that prophages would be largely optimized towards the genome of their host, but that genome composition of the extrachromosomal category would be less correlated. Particularly, as cyanophage host range breadth scales with the number of tRNA genes encoded by the virus (*Enav et al., 2012*), we expected that genome composition of viral genomes with many tRNA genes might have poor correlation to that of their host genomes, assuming that the viral-encoded tRNA genes could compensate for codon mismatches across hosts. However, these latter expectations were not met as viral and host genome composition correlations were strong regardless of the number of viral-encoded tRNA genes (*Figure 6—figure supplement 1*), which suggests that host-optimized viral genome composition may be beneficial even when the virus encodes its own tRNA genes.

Among ssDNA viruses, nucleotide composition of viral genomes was also correlated to host genomes, but less so than for dsDNA viruses (*Figure 6A*). This contrasts with a previous analysis of 500 phage genomes that did not detect any difference between dsDNA and ssDNA genomes adaptations to their host's genome (*Cardinale and Duffy, 2011*). One ssDNA viral group, the *Microviridae*, had a reduced signal for genome composition metrics except for codon usage where its signal was comparable to that of the dsDNA viruses (*Figure 6A*). Although this could indicate a bias linked to the small genome size of these viruses (around 5 kb), dsDNA viruses' genomes subsampled to similar sizes displayed a minimal signal loss (*Figure 6—figure supplement 2*), which suggests other mechanisms may be driving this lower genome composition adaptation in *Microviridae*. Another ssDNA group, the *Inoviridae* had reduced genome composition and codon usage adaptation signals. Again, because *Inoviridae* release virions without killing their hosts, it is possible that the virus is exposed to host resources over a much longer time interval, lowering the selection pressure toward transcription and translation speed and efficiency, which is the main mechanism thought to drive genome composition and codon usage adaptation of viral genomes (*Cardinale and Duffy, 2011*).

Pragmatically, to assess whether this signal could be used to predict the host of a new virus, we calculated the distance based on TNF vectors between each VirSorter curated data set sequence and the 14,977 microbial genomes. The taxonomy of the microbial genome with the lowest distance to the viral sequence (i.e., the predicted host) was then compared to the taxonomy of the actual host (i.e., the genomic data set in which the viral sequence was identified). When the host database included all host genomes, this host prediction was 99% accurate at both the family and genus level for virus–host TNF distances lower than $4.10^{-04}$, 88%/51% (family/genus level) for TNF distances ranging between $4.10^{-04}$ and $1.10^{-03}$, and 70%/37% for distances greater than $10^{-03}$ (*Table 1*). When genomes from the actual host species are excluded, the accuracy of host prediction drops slightly (95%, 83%/30%, 67%/30% for the same distance ranges), and even more when all genomes from the host genus are excluded (70% and 37% at the family level, no correct genus could be predicted in that case, and only one distance

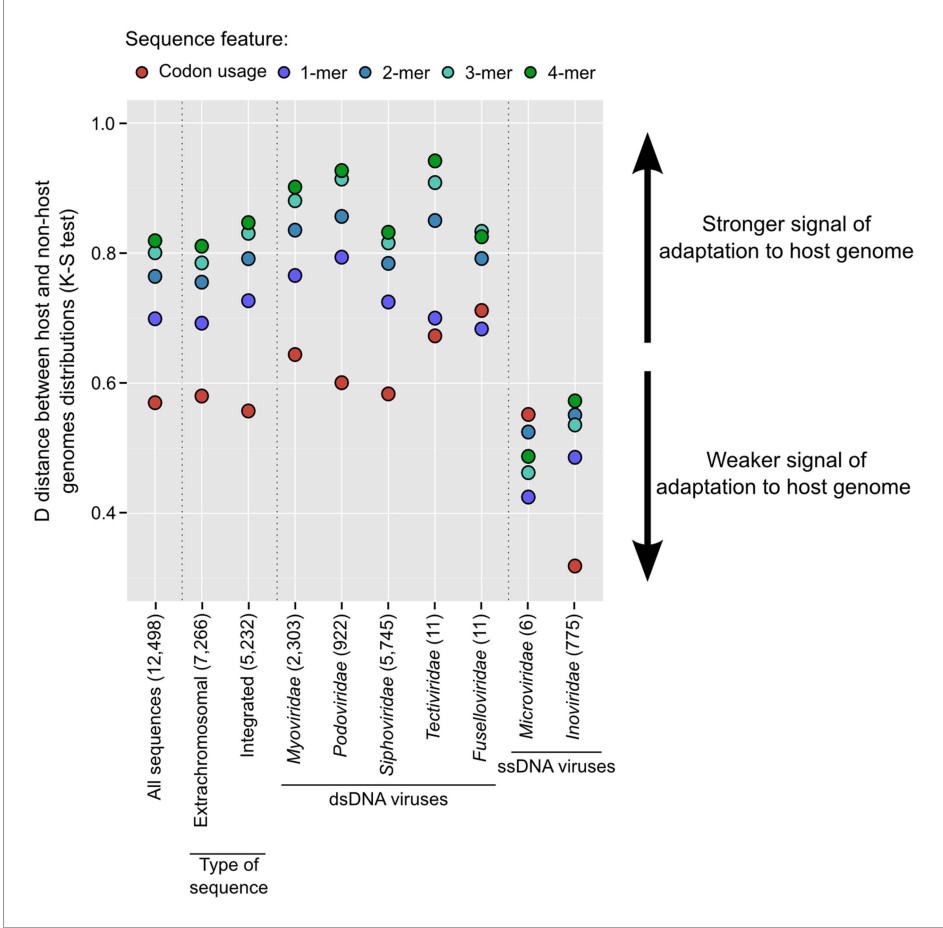

**Figure 6**. Adaptation of viral genome composition and codon usage to the host genome. K–S distances between distributions of virus–host distances and virus–non-host distances for each metrics (in color) and different subsets of the viral sequences (all sequences, by type, and by taxonomy). Only families with more than 5 genomes are displayed (although it should be noted that the VirSorter data set includes only 6 *Microviridae* sequences). The number of sequences in each category is indicated in brackets. Distributions used to compute distances are displayed in *Figure 6—figure supplement 1*.

The following figure supplements are available for figure 6:

**Figure supplement 1**. (**A**) K–S distances between distributions of virus–host distances and virus–non-host distances for each metrics (in color) and different subsets of the viral sequences (based on the number of tRNA genes detected).

**Figure supplement 2**. Distance between k-mer frequency vectors of virus genome subsamples and host genomes for *Caudovirales*.

lower than $4.10^{-04}$ was observed, *Table 1*). Hence, TNF comparison provides a promising in silico approach to link new viral genomes to hosts at different levels of accuracy within the taxonomic hierarchy when the suitable host reference genome is available.

## Data set availability

As evidenced by the improvement in virome taxonomic affiliation (*Figure 3B*), VirSorter curated data set should represent a useful reference data set for future virome studies. This data set also likely harbor novel biology beyond the global patterns of viral diversity and virus–host interactions presented in this manuscript, to be revealed through analyses targeted toward specific viral or host

**Table 1.** Accuracy of host prediction based on distance (d) between tetranucleotide frequencies of viral and microbial genomes

| | | Host order | | Host family | | Host genus | |
|---|---|---|---|---|---|---|---|
| | Predicted | Correct | Ratio (%) | Correct | Ratio (%) | Correct | Ratio (%) |
| **All reference sequences** | | | | | | | |
| $d < 4 \times 10^{-04}$ | 98 | 97 | 98.98 | 97 | 98.98 | 97 | 98.98 |
| $4 \times 10^{-04} \leq d < 1 \times 10^{-03}$ | 10,173 | 9361 | 92.02 | 8971 | 88.18 | 5261 | 51.72 |
| $1 \times 10^{-03} \leq d$ | 2508 | 1872 | 74.64 | 1757 | 70.06 | 917 | 36.56 |
| **Host species excluded** | | | | | | | |
| $d < 4 \times 10^{-04}$ | 21 | 20 | 95.24 | 20 | 95.24 | 20 | 95.24 |
| $4 \times 10^{-04} \leq d < 1 \times 10^{-03}$ | 10,003 | 9067 | 90.64 | 8372 | 83.69 | 2992 | 29.91 |
| $1 \times 10^{-03} \leq d$ | 2755 | 1981 | 71.91 | 1840 | 66.79 | 818 | 29.69 |
| **Host genus excluded** | | | | | | | |
| $d < 4 \times 10^{-04}$ | 1 | 0 | 0.00 | 0 | 0.00 | 0 | 0.00 |
| $4 \times 10^{-04} \leq d < 1 \times 10^{-03}$ | 9085 | 7303 | 80.39 | 6181 | 68.04 | 0 | 0.00 |
| $1 \times 10^{-03} \leq d$ | 3693 | 1768 | 47.87 | 1388 | 37.58 | 0 | 0.00 |

For each viral genome, the order, family, and genus of its host were predicted from the taxonomy of the closest microbial genome (based on the mean absolute difference between tetranucleotide frequency vectors) and compared to the order, family, and genus of the actual host (i.e., the taxonomy of the genome with which the virus was identified). These predictions were computed with (i) all microbial genomes, (ii) excluding specifically all genomes from the host species, and (iii) excluding all genomes from the host genus. Cases with over 75% of prediction accuracy are highlighted in gray.

subgroups. To facilitate these follow-up studies, VirSorter curated data set is made available through two complementary websites: MetaVir and iVirus. MetaVir (project 'VirSorter', data set 'VirSorter curated data set') provides an automatic annotation of each sequence, with multiple visualization tools to explore and compare genome maps, as well as multiple ways of searching the data (by host, by phage affiliation, by taxonomic or functional affiliation of predicted genes, etc) and extract a specific subset of interest (these tools are under the tab 'Contig maps'). Nucleotide sequences from the VirSorter curated data set are also hosted at iVirus, alongside the viral clusters annotation and network (as cytoscape-ready text files), the virus–host matrix, and the complete list of viral sequence predictions in the 14,977 archaeal and bacterial genomic data sets including the category 3 predictions that are not in VirSorter curated data set (http://mirrors.iplantcollaborative.org/browse/iplant/home/shared/ivirus/VirSorter_curated_dataset). Finally, a summary of the sequences and clusters is provided as *Figure 1—source data 1* and *Figure 2—source data 1*, and a Data Dryad package including all annotated genbank files from the VirSorter curated data set is available (http://dx.doi.org/10.5061/dryad.b8226; *Roux et al., 2015b*).

## Conclusion

While recent advances in high-throughput sequencing and viral metagenomics continue to expand the bounds of viral sequence space (e.g., *Reyes et al., 2012*; *Mizuno et al., 2013*; *Brum and Sullivan, 2015*), such viruses are typically unlinked to cognate hosts, severely limiting ecological and evolutionary inferences. Concurrently, emerging methods provide new virus–host linkage capabilities, but do not scale well with increasing data set size and complexity (e.g., *Andersson and Banfield, 2008*; *Tadmor et al., 2011*; *Allers et al., 2013a*; *Deng et al., 2014*). Here, the mining of publicly available microbial genomic data proved to be a useful complement to these approaches as it enables the exploration of host-linked viral diversity. The resulting viral sequences hidden in microbial genomes represent a powerful data set, increasing the number of known, host-linked viruses by an order of magnitude, with analyses of these data elucidating viral dark matter in ocean and human gut viromes, as well as augmenting our understanding of viral taxonomy, viral genome evolution, and virus–host interactions on multiple fronts. While this current

VirSorter data set remains limited by the cultivation bias inherent in the publicly available complete and draft microbial genomes, such bias will progressively be eliminated as SAGs are used to better map microbial dark matter (e.g., *Rinke et al., 2013*). Such a drastically improved map of the virosphere, together with advances in experimental approaches and theory (*Brum and Sullivan, 2015*), will help reveal the eco-evolutionary forces shaping virus–host interactions across diverse ecosystems and eventually shift our inference capability from observation to prediction.

## Materials and methods

### Application of VirSorter to public bacterial and archaeal genomes

A total of 14,977 bacterial and archaeal genomes (complete and draft) included in RefSeq and WGS databases (*Pruitt et al., 2009*) were downloaded from the NCBI ftp website in March 2014 (RefSeq release 64). The 264 new candidate phyla ('Microbial Dark Matter') SAGs' (*Rinke et al., 2013*) raw reads were downloaded from the JGI portal page and assembled with SPAdes Genome Assembler (*Bankevich et al., 2012*) (default parameters). Finally, 127 SUP05 SAGs that we previously analyzed manually were added to the cellular genome pool (*Roux et al., 2014*). This data set included 4240 complete genomes and 10,547 draft genomes (as there is no clear annotation of a genome as 'draft' or 'complete' at the NCBI, we identified as 'draft' genomes all genome projects including more than 5 different sequences, to avoid considering genomes split into different chromosomes or including one or several plasmids as 'draft').

Genomes were processed with VirSorter (*Roux et al., 2015a*) separately for each class (except for *Cyanobacteria*, SUP05 SAGs, and the Microbial Dark Matter data set that were all processed together), first using the RefSeqABVir database, and then using the Viromes database, yielding 89,301 total predicted viral sequences. Among these, 938 correspond to Enterobacteria phage PhiX174, which is used for quality control during Illumina sequencing, and were thus discarded.

### Selection of a relevant subset of viral sequences: the VirSorter data set

We focused on a subset of the putative viral sequences extracted from RefSeq, WGS and the Microbial Dark Matter and SUP05 SAGs (89,301 sequences), and targeted the active prophages and lytic virus signatures. To this end, we discarded all predictions lacking a viral hallmark gene or a viral gene enrichment (i.e., category 3 predictions, *Roux et al., 2015a*), and all prophage detections displaying viral gene enrichment only and lacking viral hallmark genes, as these are likely defective prophages for which boundaries are difficult to predict in silico and that often include bacterial genes. We next removed all linear sequences shorter than 10 kb except for sequences detected with the non-*Caudovirales* score where a threshold of 5 kb was used, as these viruses can frequently have genomes smaller than 10 kb. We also discarded all circular contigs (which should represent complete genomes) smaller than 3 kb as these are likely short repeat regions (the smallest known genome for a bacteria or archaea virus is ∼5 kb). The resulting 13,391 sequences were then manually curated to remove false positives. These false positives corresponded to defective prophages (wherein most are expected to be smaller than 10 kb), plasmid-like sequences, GTA gene clusters, and low complexity regions. In addition, this manual curation step allowed us to adjust the boundaries of some prophage predictions and/or modify the prophage vs complete viral contig automatic prediction. Consequently, 892 sequences were discarded (false-positive rate of 6.7%), leaving 12,498 curated sequences.

Among these, 7266 sequences were entirely viral (thus potentially represent lytic, chronic, or extrachromosomal lysogenic infections assembled in draft genomes), and 5232 were prophages (viral-like regions detected within a cellular genome fragment). Among the sequences detected as entirely viral, 6 were tagged in the NCBI database as bacteriophages, and 108 as plasmids. Viruses and plasmids can be difficult to distinguish, as gene exchange is known to occur between these two types of mobile genetic elements (*Leplae et al., 2010*). Here, 84 out of these 108 'plasmid' sequences displayed conclusive evidence of a viral origin as they contained viral hallmark genes (coding for terminase large subunits or major capsid proteins) and originated from draft unpublished genomes, hence likely to have been named 'plasmid' because they formed extrachromosomal circular assembly (see e.g., sequence gi:383080718 available at RefSeq). The 24 others were more ambiguous (highlighted in orange in *Figure 1—source data 1*) since the automatic annotation from NCBI did not display any viral-like gene, yet these sequences all displayed statistical viral-like gene enrichment, and as such were maintained in the VirSorter curated data set.

Finally, one additional ambiguous sequence, considered as entirely viral by VirSorter, was detected in the *Caldiserica* SAG (Caldiserica_bacterium_sp_JGI_0000059-M03_ID_3757). Even though this sequence looks indeed like a complete *Inoviridae* genome, it displayed a high level of similarity (99% identity) to the complete genome of *Delftia acidovorans* SPH-1 (gi:160361034, from coordinates 2300885 to 2307389). Such high similarity with another virus is suspicious, as well as the fact that the matching genome is *Delftia*, a bacterium known to contaminate some MDA reagents. This sequence was maintained in the VirSorter data set as there is no definite proof of the contamination, but the existence of a Caldiserica-infecting *Inoviridae* should be considered as uncertain until further evidence is available (and is displayed as such in *Figure 2—source data 2*).

## Protein and virus clustering of the VirSorter curated data set

The pool of 450,047 proteins predicted from the 12,498 viral sequences was clustered with all proteins from RefSeq and the viral metagenomes (i.e., sequences from the Viromes database) with MCL based on reciprocal best BLAST hit (threshold of 50 on score and 0.001 on e-value, *Enright et al., 2002*). Most of these sequences (423,618) could be included in 22,460 protein clusters (PCs). About a third (7742) of these PCs also contained sequences from the RefSeqABVir database, and the remainder formed new PCs.

This protein clustering was then used to cluster genomes as in Lima-Mendez et al. (*Enright et al., 2002*; *Lima-Mendez et al., 2008b*). Briefly, the number of shared PCs between each pair of sequences (either RefSeq or VirSorter) is computed, and a significance value is deduced by comparing it to an expected number of shared PCs (modeled with a hypergeometric formula taking into account the number of genes of both sequences).

We used ICCC (intracluster clustering coefficient, which estimates cluster homogeneity by measuring around each node how many of its neighbors are part of the same cluster) to determine the best inflation value (from 1.5 to 5 by 0.25 increments) and significance threshold (i.e., which minimum significance was required to draw an edge between two sequences, from 1 to 50). As expected, the number of VCs formed increased with inflation and with significance. ICCC was clearly higher with the lowest threshold in significance (sig $\geq$ 1), regardless of the inflation value used. For the lowest significance threshold, ICCC increased with inflation, usually with a first small peak around 2 and plateau around 4. These different values of inflation did not have a major impact on the clustering though, as 95–99% of pairs of sequences were clustered similarly using inflations values of 2.75, 3, 3.25, 3.5, 3.75, 4, 4.25, 4.5, 4.75, or 5. We eventually settled for the combination yielding the highest ICCC: a significance threshold of 1 and inflation of 4 (*Figure 2—figure supplement 2*).

## Taxonomic and functional affiliation of sequences and VCs

Taxonomic affiliation of sequences was based on hits to the RefSeqABVir database. Each profile in the database was first affiliated based on the origin of its members, with a 75% majority rule: at each taxonomic level, a profile is affiliated to a taxon if more than 75% of the profile sequences are affiliated to this taxon. Then, for each of the 12,498 viral sequences identified by VirSorter, a set of relevant hits was selected: (i) first the profile with the best hit across all genes along the sequence, and (ii) the best hit from other genes with a score close to this 'absolute' best hit in the sequence (>75% of the score of the first best hit). The sequence was then affiliated to the Lowest Common Ancestor (LCA) of this set of relevant hits. Hence, a predicted protein will only be affiliated if pointing toward sequences or profiles typical of a viral group, and a sequence detected by VirSorter will only be affiliated if its best hits are consistent. Functional affiliation for each PC was based on the comparison of its members (predicted proteins) with PFAM (v. 27, threshold of 50 on score). VCs were affiliated based on its members affiliations if >75% were consistent.

For the detection of new genera in the VCs, we chose to ignoring the 79 VCs that lacked large (>30 kb) genome sequences. This 30 kb threshold is conservative as it avoids considering short genome fragments as new genera but would also overlook small non-circular viral genomes (such as some *Tectiviridae*). However, because the latter comprise a minority (~0.1% of 12,498 sequences) of the VirSorter data set (*Figure 2*), we chose to retain the larger, more conservative threshold.

The 7 short circular sequences from *Bacteroidia* only detected with the Viromes database (gi 319430465, 298484481, 329959038, 423221334, 423242675, 423298785, 345651594) were targeted for further examination. Hits to PFAM domains could be found on two proteins: a relaxase (PF03432.9, score ~170), and one replication initiator protein (PF01051.16, score ~80). Genome organization was

compared with Easyfig (*Sullivan et al., 2011*) after aligning all genomes to the same starting point (one base before the start of the Rep-domain protein). Recruitment plots of virome contigs (extracted from *Kim et al., 2011*; *Minot et al., 2012*) were generated with ggplot2 and based on blastn comparison.

## Host range and co-infection

The virus–host network was assessed considering only VCs with more than 10 sequences. Hosts were grouped at the class level. The modularity Q value of the virus–host matrix was computed with the lp-BRIM module in R software (https://github.com/tpoisot/lp-brim). The virus–host matrix had a modularity of 0.45. The same index computed from 99 randomly permuted matrices ranged from 0.02 to 0.17, with an average of 0.07.

Co-infection was defined as the detection of several distinct viruses in one genome project (one complete genome or one SAG). However, different viral contigs in a single draft genome could also originate from a single viral genome mis-assembled in several different contigs. This will be especially true for *Caudovirales* that are the most detected viruses as well as the ones with the largest genomes. To limit the over-estimation of co-infection due to mis-assembled *Caudovirales* genomes, co-infection was only considered in the cases where multiple copies of the large subunit of the terminase were detected, because this gene is present in single copy in *Caudovirales* genomes, and usually detected even in new viruses (due to a high level of sequence conservation).

## Evaluation of virus–host genome adaptation

Relative frequencies of k-mers (mono-, di-, tri-, and tetra-nucleotide) were computed with Jellyfish (*Marçais and Kingsford, 2011*) for every VirSorter sequence and every bacterial and archaeal genome initially mined. Mean absolute error (i.e. average of absolute differences) between k-mer frequency vectors were then computed with an in-house perl script for each pair of VirSorter sequence and cellular genome, and used as a distance metric between viruses and putative hosts. For each VirSorter sequence, a set of distances that included its host (i.e., the genome with which the sequence was initially associated) alongside 10 randomly selected sequences from the same genus, the same family, and a different order than the host were factored into in the distance distribution (*Figure 6—figure supplement 1*).

Codon usage adaptation was evaluated with cusp and cai from the European Molecular Biology Open Software Suite (EMBOSS, *Rice and Longden, 2000*). First, codon usage bias of each bacterial and archaeal genome was computed. Then, the codon usage adaptation index (cai) was calculated for each gene between VirSorter sequences and cellular genomes. The global distribution displays the average (across genes) adaptation index for each VirSorter sequence and (as for the k-mer distances) a subset of cellular genomes including its host and 10 randomly selected sequences from respectively the same genus, the same family, and a different order than the actual host. Function-specific codon usage bias was based on the gene-by-gene adaptation between each VirSorter sequence and its host.

For each category studied, the distance between distribution of distances to host genome (in red on *Figure 5—figure supplement 1*) and distribution of distances to non-host genomes (in purple on *Figure 5—figure supplement 1*) was evaluated with a Kolmogorov–Smirnov (K–S) statistic. The codon usage adaptation indexes for the different functional categories were compared to the 'other functions' values with a Wilcoxon signed-rank test to detect categories with statistically different averages. Both statistics were computed with R software.

To evaluate the effect of small genome size on distance between k-mer frequencies, a sub-sample of 1000 *Caudovirales* was randomly taken at different sizes (from 2000 to 20,000 bp), and the same procedure as for complete sequences was used to determine the distance between host and non-host distributions of k-mer distances. Even though the signal was slightly less strong for shorter fragments, this simulation indicates that genome size is not the only factor that could explain such low viral–host genome adaptation for ssDNA viruses.

The prediction of the host taxonomy for each viral sequence was based on the microbial genome with the lowest tetramer frequency distance to the viral sequence. A prediction was considered as 'correct' when this closest microbial genome taxonomy was the same as the original genome in which the viral sequence was detected. This prediction was computed using (i) all microbial genomes, (ii) only genomes from a different species than the actual host (i.e., the genome in which the viral sequence was originally detected), and (iii) only genomes from a different genus than the actual host.

## Estimation of virome affiliation improvement and prophage assembly efficiency

Protein sequences predicted from the POV (*Hurwitz and Sullivan, 2013*), TOV (*Brum et al., 2015*), and human gut viromes (*Minot et al., 2012*) data sets were compared to RefSeqABVir (Jan. 2014) using BLAST (blastp, threshold of 50 on bit score and 0.001 on e-value). Those proteins that did not affiliate at >50 bit score and <0.001 e-value thresholds were considered 'unclassified' and then used as queries in a secondary BLAST (blastp with the same thresholds) against the predicted proteins from the VirSorter curated data set. Any unclassified proteins matching the VirSorter data set better were considered newly affiliated.

To evaluate the efficiency of prophage assembly, we simulated genome sequencing from 23 bacterial genomes with identified prophages (NC_000907 NC_000913 NC_000964 NC_002570 NC_002655 NC_002662 NC_002695 NC_002935 NC_003030 NC_003212 NC_003295 NC_003366 NC_003997 NC_004070 NC_004307 NC_004310 NC_004431 NC_004557 NC_004567 NC_004668 NC_004722 NC_005085 NC_005362). NeSSM (*Jia et al., 2013*) was used to simulated HiSeq Illumina reads (100 bp paired-ends) with a coverage of the prophage region varying between 5×, 25×, 50×, 75×, and 100×. Reads were then assembled with Idba_ud (*Peng et al., 2012*), and viral contigs were predicted with VirSorter (*Roux et al., 2015a*). On the 481 contigs larger than 30 kb detected as viral by VirSorter, 11 were considered as 'entirely viral' even though these originated from integrated prophages, resulting in a 'false-positive' ratio of integrated prophages wrongly considered as extrachromosomal viral genomes of 2.3% for contigs of 30 kb and more. As could be expected, this same 'false-positive' ratio was higher for smaller contigs (12.06% for contigs <20 kb, and 22.81% for contigs <10 kb), so that we considered the origin of these small contigs as 'undetermined', since they may come from integrated prophages or extrachromosomal genomes.

All scripts used in this study are available on the TMPL wiki as a zip package: http://tmpl.arizona.edu/dokuwiki/doku.php?id=bioinformatics:scripts:vsb and on github: http://github.com/simroux/virsorter-curated-dataset-scripts-package.

## Acknowledgements

We thank Natalie Solonenko and Sheri Floge and TMPL members for their comments on the manuscript. This work was performed under the auspices of the Gordon and Betty Moore Foundation (#3790) through grants awarded to MBS and the Natural Sciences and Engineering Research Council (NSERC) of Canada, Canada Foundation for Innovation (CFI), the Canadian Institute for Advanced Research (CIFAR), and the Tula Foundation funded Centre for Microbial Diversity and Evolution, G Unger Vetlesen and Ambrose Monell Foundation through grants awarded to SJH. The work conducted by the U.S. Department of Energy Joint Genome Institute, a DOE Office of Science User Facility, is supported under Contract No. DE-AC02-05CH11231.

## Additional information

### Funding

| Funder | Grant reference | Author |
| --- | --- | --- |
| Gordon and Betty Moore Foundation | 3790 | Matthew B Sullivan |
| Natural Sciences and Engineering Research Council of Canada (Conseil de Recherches en Sciences Naturelles et en Génie du Canada) | | Steven J Hallam |
| Canada Foundation for Innovation (Fondation canadienne pour l'innovation) | | Steven J Hallam |
| Canadian Institute for Advanced Research (L'Institut Canadien de Recherches Avancées) | | Steven J Hallam |
| Tula Foundation | | Steven J Hallam |

| Funder | Grant reference | Author |
| --- | --- | --- |
| Ambrose Monell Foundation | | Steven J Hallam |
| G. Unger Vetlesen Foundation | | Steven J Hallam |
| U.S. Department of Energy (Department of Energy) | Joint Genome Institute (DE-AC02-05CH11231) | Tanja Woyke |

The funders had no role in study design, data collection and interpretation, or the decision to submit the work for publication.

## Author contributions

SR, MBS, Conception and design, Acquisition of data, Analysis and interpretation of data, Drafting or revising the article; SJH, TW, Conception and design, Drafting or revising the article

## Additional files

### Major dataset

The following dataset was generated:

| Author(s) | Year | Dataset title | Dataset ID and/or URL | Database, license, and accessibility information |
| --- | --- | --- | --- | --- |
| Roux S, Hallam SJ, Woyke T, Sullivan MB | 2015 | Data from: Viral dark matter and virus-host interactions resolved from publicly available microbial genomes | http://datadryad.org/resource/doi:10.5061/dryad.b8226 | Available at Dryad Digital Repository under a CC0 Public Domain Dedication. |

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
