## [Decision Letter]

[Editors’ note: this article was originally rejected after discussions between the reviewers, but the authors were invited to resubmit after an appeal against the decision.]

Thank you for choosing to send your work entitled “Viral dark matter and virus-host interactions resolved from publicly available microbial genomes” for consideration at *eLife*. Your full submission has been evaluated by Diethard Tautz (Senior editor) and three peer reviewers, one of whom is a member of our Board of Reviewing Editors, and the decision was reached after discussions between the reviewers. One of the three reviewers, Ken Stedmann, has agreed to share his identity.

All reviewers agreed that virus diversity and ecology are very important topics that deserve more attention and new approaches. The large set of putative viral sequences is impressive and the patterns of host association are intriguing, but we felt that the analysis didn't deliver much novel insight into the evolution of “viral dark matter”. A more in depth analysis covering multiple scales of evolution would be necessary to make significant progress in this direction. As it stands, the manuscript describes in broad terms a data set generated with a tool that is supposed to be published elsewhere. The usefulness of the data set as a resource for others remains limited without a feature rich data base that allows convenient exploration and access. Hence we don't feel that your manuscript is appropriate for publication as a Research article in *eLife*.

Reviewer #1:

Roux and colleagues analyze a large set of putative viral sequences mined from published bacterial and archaeal genomes using a software that is described in a manuscript that is currently under review elsewhere (and provided to the reviewers). The method seems sound and I think the majority of the reported viral sequences are genuine. The authors use this large set (10 fold larger than existing data bases) to investigate patterns of host adaptation, host range, and virus taxonomy.

The main results reported are:

(i) > 12000 sequences fall into ∼600 clusters, half of which contain known viruses;

(ii) Viruses are well adapted to their hosts, across virus types and proteins;

(iii) Viruses are mostly host specific and virus/hosts define modules.

These results make sense and are mostly expected, the novel element here is to be able to do it on a massive scale. I have a couple of other comments/criticisms:

1) Only the more reliable predictions were used. How do things change when less stringent criteria are used?

2) Is there a sense of saturation: if one had done the study a few years ago with fewer genomes, how many clusters would have been found? Are some parts of the bacterial world exhausted, where do the new sequences come from?

3) What can be learned from this for future efforts to detect viruses? How should sequencing be targeted?

4) Coinfection: the number of viruses per genome seem compatible with random. What can really be learned from this? Does this reflect the number of concomitant infections, or the number of genomes deposited in this genome in the past (like endogenous retroviruses in mammals)?

5) How are others supposed to use this? A data set of this size needs tools to analyze. Are the authors going to develop a data base with interactive views etc?

6) This is a computational study. I expect the scripts and code to be deposited.

Reviewer #2:

In this manuscript the authors describe an approach to increase our knowledge about prokaryotic viruses (phages) by mining prokaryotic genomes available in public repositories. It is an excellent idea that makes a lot of sense and is badly needed to increase the knowledge about the sequence space of phages presently very biased and incomplete. It can also provide a great contribution into one of the conundrums of phage biology, the infection range without the bias of culture. However, I have mixed feelings about this manuscript. On the one hand it is very comprehensive including all genomes in repositories (including many draft genomes) but the results are a bit disappointing and provide very little novelty. That the pattern of infection at large phylogenetic scale will be modular was largely expected from classical work with cultures. But the most relevant question is whether at short phylogenetic distances is nested what is left unanswered. Maybe a problem that is general to these “big data” analysis is the gross level of detail. I wonder why the authors do not provide analysis at the fine resolution level i.e. phages detected within a single species or genus. At the broad level analysed here most of the results are very predictable from classic approaches. The use of draft genomes and the possibility of discriminating plasmids from phages is another question that is left untouched in both this manuscript and the previous submission. There is a gradient in nature between infective phages and conjugative elements and establishing the borderline might be risky.

In summary I missed some more fine grained analysis of examples in this big data approach.

Reviewer #3:

I find this to be a very well-written report of the application of a new bioinformatic tool (VirSorter) developed by some of the authors. This tool has been applied to data mining of the available and rapidly growing genomic datasets and thereby has increased the number of putative (mostly partial) viral genomes by ten-fold. Due to association with both known genomes and SAG genomes from known sources, the analysis allowed identification of potential viruses in hosts for which no known viruses are currently available. This is clearly a boon to researchers working on these organisms. I find the tetranucleotide analysis of viral genomes in order to possibly identify hosts for these viruses to be particularly attractive and plan on using it in my own research.

I am not convinced of the premise stated in the title and Abstract that this analysis provides much insight into viral dark matter or virus-host interactions. This tool enables further investigations that would allow that insight, which would otherwise be extremely difficult if not unfeasible.

I wonder if the manuscript describing the development and testing of the tool (which was submitted together with this manuscript) could be combined into one manuscript.

[Editors’ note: what now follows is the decision letter after the authors submitted for further consideration.]

Thank you for resubmitting your work entitled “Viral dark matter and virus-host interactions resolved from publicly available microbial genomes” for further consideration as a Tools and resources article at *eLife*. Your revised article has been favorably evaluated by Diethard Tautz (Senior editor) and a Reviewing editor. The manuscript has been improved but there are some remaining issues that need to be addressed before acceptance, as outlined below:

1) Data availability: given that this is now being considered as a Tools and resources article, we feel that the data availability section should be more prominent. We suggest to move the availability section up (possibly before the conclusions) and provide a little bit more detail. iVirus.us itself seems like a rather hollow shell – clicking on data access yields a 502 bad gateway error. As far as I can tell, everything happens within the discovery environment of iPlant for which registration is necessary. Please elaborate a little bit. The MetaVir environment seems useful, but some of the MetaVir analysis haven't completed yet. In addition, we think that the “richly annotated genbank files” (promised in the rebuttal letter) should be made available not only on the author's website but uploaded to a big-data repository such as data dryad.

2) It seems the authors have misunderstood the request for making the scripts available. We were not asking for the VirSorter scripts, but the scripts that analyze the VirSorter data set to produce figures and results of the paper. Those scripts provide the most accurate description of the methods, and in the interest of reproducibility, they should – whenever possible – be made available. The preferred place would be a separate GitHub repository.

3) Host association figure: We continue to be underwhelmed by this figure. There are lots of lines which clearly fall into a handful of modules, but within these modules it is pretty hard to see what is going on. Maybe a two-way clustering would be more insightful. Consider a distance matrix d_ij, where d_ij is the fraction of sequences in viral cluster (VC) i that come from genomes of host phylum j, maybe normalized for the abundance of genomes from phylum j. Then cluster this matrix both by VC and host, similar to RNA-seq being clustered by gene and tissue. The modules should show up as blocks on the diagonal, while promiscuous affiliations are off-diagonal terms. What exactly the distance matrix d_ij should be requires some thought and there are probably better choices then this proposal. But if something like this would work out, it could be more informative than the current figure. Keeping one as a supplement of the other could be a good solution.

---

## [Author Response]

*All reviewers agreed that virus diversity and ecology are very important topics that deserve more attention and new approaches. The large set of putative viral sequences is impressive and the patterns of host association are intriguing, but we felt that the analysis didn't deliver much novel insight into the evolution of* “*viral dark matter*”*. A more in depth analysis covering multiple scales of evolution would be necessary to make significant progress in this direction. As it stands, the manuscript describes in broad terms a data set generated with a tool that is supposed to be published elsewhere. The usefulness of the data set as a resource for others remains limited without a feature rich data base that allows convenient exploration and access. Hence we don't feel that your manuscript is appropriate for publication as a Research article in* eLife.

We can appreciate that a manuscript introducing 12,498 new phage genomes (whole and large fragments) leaves a feeling of unfinished business no matter how it is written. Seeing these reviews also help us see that we failed to really start out with a quantitative metric of “impact” as to us the scale alone (augmenting available phage genome sequences by an order of magnitude) was a closed case. This is because the last decade has seen microbial ecology transformed by large scale datasets including the Global Ocean Survey microbial metagenomics dataset (Rusch et al. PLoS Biology, 2007) and the first viral metagenomic dataset (Angly et al. PLoS Biology, 2006) – papers which have 1383 and 613 citations, respectively. At the same time, viral ecology is paralyzed by the dominance of “unknowns” in metagenomics studies as commonly 63–93% of new viral metagenomic reads are new to science, presumably because we only have just over a thousand phage genomes and they derive largely (85%) from only 3 of 45 bacterial phyla.

How much of a difference will our 12,498 host-associated phage genomes improve the situation? A new analysis we include here shows that they as much as double the number of affiliated proteins for some environmental viromes (∼35% for seawater viromes vs ∼100% for human gut virome; see Figure 7). Thus we hope this more clearly emphasizes how single study’s dataset alone will be foundational for future ecology studies seeking to “see” viruses in microbial datasets and to affiliate viruses in viral datasets. These new results were added to the revised manuscript (text and new Figure 3).

Author response image 1.Improvement in the proportion of affiliated genes from viromes with VirSorter dataset.Predicted genes from the Pacific Ocean Viromes (36), Tara Ocean Viromes (8) and Human Gut Viromes (Minot et al., 2013) were compared to RefSeqVirus (May 2015) and the 12.5k VirSorter dataset (BLASTp, threshold of 50 on bit score and 0.001 on e-value). Predicted proteins affiliated to VirSorter (in blue) did not display any significant similarity to a RefSeq virus, but can now be associated with a phage and a host through the VirSorter database.**DOI:**
http://dx.doi.org/10.7554/eLife.08490.024

As well, we can see that we failed to clearly articulate to the reviewers the specific biological advances made in this manuscript. To summarize these advances, we list the major advances here, any single of which, I would argue, could be the sole focus of a strong, top-tier manuscript.

1) The amount of viral signal in publicly deposited genomes (12.5k highly confident viral sequences in 15k bacteria and archaea genomes) is unexpectedly high since we focused our analysis on “active” infections by excluding fragmented genomes likely to be defective or decayed prophages.

2) This study is the first to attempt to quantify the lesser studied types of viruses and finds viral genomes not integrated in the host genome to be rather abundant (>1k sequences were identifiable, subsection “New viruses detected in public microbial genomic datasets with VirSorter”). These could represent extrachromosomal prophages, chronic, or “cryptic” lytic viruses (i.e. lytic viruses that goes unnoticed in a culture), all infection types that are understudied and with unknown and likely underestimated ecological impacts.

3) Genome-based clustering analyses revealed that approximately half of the observed viral clusters in the VirSorter dataset lacked known reference genomes (subsection “264 new putative viral genera identified through genome-based network clustering”, last paragraph). Obtaining complete or near-complete genomes and documenting the host range for these new groups is critical for mapping the virosphere, especially because while other approaches (e.g., viral metagenomics) can help identify non-cultivated viral diversity, these lack this host association. Highlightable “firsts” here include the first viral genomes for 9 bacterial phyla (subsection “Long-term evolutionary patterns of bacterial and archaeal virus genomes” and Table 1, see also Figure 8), which is about as much from all published literature to date, as well as a new *Bacteroides* virus, unrelated to any virus previously described that likely represents a new viral order (subsection “New viruses detected in public microbial genomic datasets with VirSorter”, third paragraph).

Author response image 2.Viral sequences distribution of RefSeq and VirSorter dataset.For each host group, a circle proportional to the number of viral genomes available is noted in red for RefSeq and blue for VirSorter. Hosts for which no RefSeq references were available are highlighted in bold.**DOI:**
http://dx.doi.org/10.7554/eLife.08490.025

4) The fraction of microbes in any environment that are co-infection by more than one virus remains a fundamental, yet largely (completely?) unknown number in any environment. Here we show that co-infection is common (∼50% of cells are co-infected, l. 242) and many of these co-infections are by more than one type of virus. Such co-infections likely have far-reaching implications for viral genome evolution, as they provide opportunity for gene exchange, so quantifying the co-infection frequency across viral groups offers insight into how genomically promiscuous one viral group might be relative to another.

5) While not completely novel observations, we also perform analyses that confirm, with a much larger dataset, prior work in key areas that are desperately needed in viral taxonomy and ecology. First, genome network analysis helps classify new viruses in a robust genome-based taxonomic framework that is largely consistent with accepted ICTV taxonomy (subsection “264 new putative viral genera identified through genome-based network clustering”). Second, leveraging this unprecedentedly large-scale, host-associated viral genomic dataset, we show that tetranucleotide frequency distance is a surprisingly robust predictor of the host of most viruses. Again, while not novel knowledge, working at this scale we added an effort to quantify the probabilistic value of these predictions across multiple host phyla, as well as compared the performance of tetranucleotide frequency to 4 other sequence composition based metrics to help provide strong guidance to researchers on using this metric. Perhaps this is why in spite of the idea being in the literature for some time, reviewer #3 notes: “I find the tetranucleotide analysis of viral genomes in order to possibly identify hosts for these viruses to be particularly attractive and plan on using it in my own research*.*” Third, the modular pattern of a global virus-host network was indeed predicted by theoretical models, although the only study of comparable size observing a modular virus-host network (Flores et al., ISMEJ, 2011) was based on plaque formation on host cultures where genetic diversity was unknown. Here, we validate the modularity with an unprecedentedly large-scale dataset that includes microbes spanning 18 phyla, and add information about the level of taxonomy at which the virus-host network is modular (as we expect it to become nested at one point, near the “tip of the tree”). Fourth, the dominance of Caudovirales in the dataset, as well as the clear separation between DNA and RNA viruses as well as Archaeal and Bacterial viruses were all expected based on the previous knowledge on viral diversity, and so these findings are largely only confirmatory.

To better emphasize these results, we added a new figure displaying more clearly how the curated VirSorter dataset expands the range of known viruses (Figure 1), and re-organizing the manuscript so that the first three subparts of Results section are now entirely dedicated to the exploration of this new diversity (subsection “New viruses detected in public microbial genomic datasets with VirSorter”). The questions of viral classification through genome-based network, virus-host interactions and adaptations are then addressed in three more subsections (“Long-term evolutionary patterns of bacterial and archaeal virus genomes“, Global virus–host network is confirmed as modular” and “Virus–host adaptation signals detected at the genome composition and codon usage level”). We hope that this new organization brings more balance to the manuscript and helps to better introduce the dataset before actually switching to secondary analyses.

Another issue that the reviewers had was that there was the perception of a lack of depth in the manuscript. We acknowledge that the format of the manuscript follows a style in which only global patterns are presented. This is similar to how Rinke et al*.* (2013 Nature) handled their explorations of microbial dark matter – and is a common strategy for getting such big datasets out to specialists for follow-on analyses. Such follow-up studies will undoubtedly be extremely interesting and critical for the field. However, we chose to play to the strengths of the data and consider only global-scale stories.

Notably, *eLife* recently published a manuscript describing comparative phage genomics of 627 mycobacteriophage genomes (Whole genome comparison of a large collection of mycobacteriophages reveals a continuum of phage genetic diversity, Pope et al., 2015. *eLife*). The major findings in the manuscript (e.g., phage genomes are part of a genetic continuum) are consistent with and redundant to the findings from at least 5 previously published papers by the same group (Hendrix et al., PNAS, 1998; Pedulla et al., Cell, 2003; Jacobs-Sera et al., Virology, 2012; Cresawn et al., PLoS One, 2015) yet the value of such a large-scale dataset is paramount currently for future studies of the ecology, evolution and genomics of phages, which is presumably why the study was accepted at *eLife*.

Finally, the last two criticisms in the editor’s summary were as follows:

1) *“the manuscript describes in broad terms a data set generated with a tool that is supposed to be published elsewhere.”* The manuscript describing the methodological details of the tool is now published VirSorter: mining viral signal from microbial genomic data”, S Roux, F Enault, BL Hurwitz, MB Sullivan, PeerJ 3, e985.

2) *“The usefulness of the data set as a resource for others remains limited without a feature rich data base that allows convenient exploration and access.”* We strongly agree that providing convenient and enriched access to data and tools is crucial for researchers, as can be seen by the previous projects of the lead author who developed and maintain one of the only viral metagenomic databases and analysis tools publicly available (MetaVir), and the senior author’s laboratory which has been building iVirus on the back of the NSF-funded iPlant Cyberinfrastructure in spite of a lack of funding for the project. In fact, both projects are unfunded, yet we maintain and/or develop the efforts as they represent our commitment to getting the data and tools into researchers hands to enable them to better “see” the viruses in their datasets. Although we did not emphasize this in the manuscript, a mistake we would correct in a revised manuscript, we intend to make the dataset available on these two complementary websites (MetaVir and iVirus). MetaVir provides an automatic annotation of each sequence, with multiple visualization tools to explore and compare genome maps, as well as multiple ways of searching the data (by host, by phage affiliation, by gene taxonomic or functional affiliation, by size, etc) and extract a specific subset of interest. iVirus offers optimized data repository features as well as numerous analytical tools for comparative genome analyses and metagenomic fragment recruitment analyses (BLAST, bwa and bowtie2 read aligners, multiple flavors of functional gene annotation, phylogenetic tree building pipelines, etc.). Finally, a summary of the sequences and clusters is provided as supplementary files, and both the raw data and richly annotated sequences (genbank file format, including taxonomic and functional affiliation of all genes) will be available to download on Sullivan’s publications webpage should the paper be accepted – just as we do for other datasets of community interest (e.g., the Pacific Ocean Viromes and the Tara Oceans Viromes datasets). The information about the dataset availability was added to the Material and methods (subsection “Dataset and script availability”).

In summary, we would argue that there is no greater challenge to exploring the ecology and evolution of viral communities in diverse ecosystems (e.g., oceans, soils, humans) than the lack of reference genomes that cause dominance by ‘viral dark matter’. Above we have tried to more carefully articulate the major advances this study makes, and we emphasize that the reviewers also noted the quality of the work and its relevance for the field (Reviewer 2: *“It is an excellent idea that makes a lot of sense and is badly needed to increase the knowledge about the sequence space of phages presently very biased and incomplete.”*, Reviewer 3: *“This is clearly a boon to researchers working on these organisms.”*). We hope that the new figures (Figures 1, 2 and 3), the added results and new organization of the manuscript helped to bring out how valuable VirSorter curated dataset is, and what insights into virus-host interactions were obtained. Please find below a point-by-point response to the reviewers comments.

Reviewer #1:

*Roux and colleagues analyze a large set of putative viral sequences mined from published bacterial and archaeal genomes using a software that is described in a manuscript that is currently under review elsewhere (and provided to the reviewers). The method seems sound and I think the majority of the reported viral sequences are genuine. The authors use this large set (10 fold larger than existing data bases) to investigate patterns of host adaptation, host range, and virus taxonomy*.

The main results reported are:

*(i) > 12000 sequences fall into ∼600 clusters, half of which contain known viruses*;

*(ii) Viruses are well adapted to their hosts, across virus types and proteins*;

*(iii) Viruses are mostly host specific and virus/hosts define modules*.

*These results make sense and are mostly expected, the novel element here is to be able to do it on a massive scale*.

We acknowledge that some results mostly confirm what could be predicted based on smaller scale studies, but we feel that we have made significant new discoveries and phenomenological observations in exploring the global scale patterns in this dataset. We hope that our efforts above (response to editor’s summary) now better articulate the specific advances made in this manuscript.

*I have a couple of other comments/criticisms*:

*1) Only the more reliable predictions were used*. *How do things change when less stringent criteria are used?*

We appreciate the suggestion of including the category 3 predictions (∼90K sequences) in our analyses. During the data exploration phase of preparing this manuscript, we examined the category 3 predictions but found it to be of mixed use since we were focused on viral sequence space in this manuscript. We went into some detail about this in the “tool” manuscript in PeerJ*,* but also here explicitly caution the reader about the value of category 3 predictions. Specifically, “we discarded all predictions lacking a viral hallmark gene or a viral gene enrichment […] as these are likely defective prophages for which boundaries are difficult to predict in silico and that often include bacterial genes” (subsection “Selection of a relevant subset of viral sequences: the VirSorter dataset”). While we were focused here on the higher confidence viral genome sequences (category 1 and 2 predictions), the category 3 predictions are of great value to specialists interested in defective prophages, mobile elements or microbial genomic islands. We hope with this added context that you can appreciate our decision to focus in this way and yet also make the category 3 predictions available through this study since they could be of value for follow-on work.

2) Is there a sense of saturation: if one had done the study a few years ago with fewer genomes, how many clusters would have been found? Are some parts of the bacterial world exhausted, where do the new sequences come from?

Because this study leverages 15K publicly available microbial genomes, it is not an ideal dataset from which to draw conclusions about saturation. Notably, however, we saw that even well studied groups, such as *Gammaproteobacteria* and *Bacilli,* do not appear saturated as new VCs (i.e., those lacking a RefSeq reference) were detected here too (new Figure 2). We do see this as an ideal question to approach in the future using VirSorter – once the floods of SAGs data are available as these sequences will better span the microbial tree of life and provide context for both lytically infecting and cell-associated (prophage, extrachromosomal, chronic, etc.) infecting viruses.

3) What can be learned from this for future efforts to detect viruses? How should sequencing be targeted?

This is a question each individual researcher will need to answer based upon their particular research question of interest – are you interested in capturing sequence breadth or depth? It’s an age-old trade-off and one we do not answer well here since even this scale of data is not very deep in any one category yet since we leveraged public data rather than develop an explicit experimental sampling strategy.

*4) Coinfection: the number of viruses per genome seem compatible with random. What can really be learned from this? Does this reflect the number of concomitant infections*, *or the number of genomes deposited in this genome in the past (like endogenous retroviruses in mammals)?*

Unfortunately, discerning genomes previously deposited in the host genome from active viral infections in silico is nearly impossible. We conservatively focus on sequences that are likely to be active and not past infections as we only consider prophages that included the capsid-associated genes (viral hallmark genes). Thus, the 12.5k sequences likely underestimate the total number of viruses in the dataset (since we miss those with unrecognizable capsid genes), but should conservatively identify active infections that represent some combination of lytic infections, prophages and chronic infections. We added a discussion about active viruses (subsection “VirSorter curated dataset includes extrachromosomal genomes and improves virome affiliation”).

*5) How are others supposed to use this? A data set of this size needs tools to analyze. Are the authors going to develop a data base with interactive views etc*.*?*

We carefully considered for some time how best to make these data available and in the end chose to make it available through the iPlant Cyber infrastructure, which allows to easily share large sequences datasets, and the MetaVir web server, which generates automatically annotated contig maps searchable by function, taxonomy, or host taxonomy (details in the response to editor's summary above). Notably, all viral genomes are made available in a fully-annotated genbank format that could be utilized by researches in any number of genome browsers (e.g., Artemis) for follow-on analytics using the tool of choice. It is beyond the scope of this manuscript or our lab to develop interactive data interrogation tools for these data as these efforts are often large-scale projects (e.g., iPlant is a $100M NSF Center, KBase is $100M DOE Center).

*6) This is a computational study. I expect the scripts and code to be deposited*.

We agree, and apologize for not displaying this clearly in the manuscript. All of our code was previously made publicly available through GitHub and a community-available version is implemented through the iPlant Cyberinfrastructure. These details were in the prior, PeerJ publication that describes the VirSorter tool, but we now also point readers to these details in the current manuscript (subsection “Dataset and script availability”).

Reviewer #2:

*In this manuscript the authors describe an approach to increase our knowledge about prokaryotic viruses (phages) by mining prokaryotic genomes available in public repositories. It is an excellent idea that makes a lot of sense and is badly needed to increase the knowledge about the sequence space of phages presently very biased and incomplete. It can also provide a great contribution into one of the conundrums of phage biology, the infection range without the bias of culture. However, I have mixed feelings about this manuscript. On the one hand it is very comprehensive including all genomes in repositories (including many draft genomes) but the results are a bit disappointing and provide very little novelty. That the pattern of infection at large phylogenetic scale will be modular was largely expected from classical work with cultures. But the most relevant question is whether at short phylogenetic distances is nested what is left unanswered. Maybe a problem that is general to these* “*big data*” *analysis is the gross level of detail. I wonder why the authors do not provide analysis at the fine resolution level i.e. phages detected within a single species or genus. At the broad level analysed here most of the results are very predictable from classic approaches. The use of draft genomes and the possibility of discriminating plasmids from phages is another question that is left untouched in both this manuscript and the previous submission. There is a gradient in nature between infective phages and conjugative elements and establishing the borderline might be risky*.

In summary I missed some more fine grained analysis of examples in this big data approach.

We thank the reviewer for these kind words. Indeed, these more detailed analyses would likely be extremely interesting, however the density of the current manuscript (see the reply to editor's summary) hardly allows for the addition of more results, which would also lead to additional Introduction and Discussion. In the response to the editor’s summary above, we describe our rationale for why we hope to keep the manuscript focused on the big picture or global-scale analyses.

Reviewer #3:

I find this to be a very well-written report of the application of a new bioinformatic tool (VirSorter) developed by some of the authors. This tool has been applied to data mining of the available and rapidly growing genomic datasets and thereby has increased the number of putative (mostly partial) viral genomes by ten-fold. Due to association with both known genomes and SAG genomes from known sources, the analysis allowed identification of potential viruses in hosts for which no known viruses are currently available. This is clearly a boon to researchers working on these organisms. I find the tetranucleotide analysis of viral genomes in order to possibly identify hosts for these viruses to be particularly attractive and plan on using it in my own research.

*I am not convinced of the premise stated in the title and Abstract that this analysis provides much insight into viral dark matter or virus-host interactions. This tool enables further investigations that would allow that insight, which would otherwise be extremely difficult if not unfeasible*.

We thank the reviewer for the kind words. Although we agree that the tool enables exciting potential follow-up investigations, we still consider that the description of such a vast dataset, that includes (as noted by the reviewer) potential viruses for host groups with no currently isolated virus, is akin to taking one (giant) step into the viral dark matter. Notably, using this host-associated viral sequences as a complementary database doubled the ratio of affiliated genes from human gut viromes (see Figure 7). The description of the different viral clusters linked to these new viruses and associated with specific host groups is for us what we consider as new insights into viral dark matter and virus -host interactions. Clearly we failed to articulate those advances in the submitted manuscript, but hope that our response to the editor’s summary above helps more clearly make our case. We hope this revised manuscript is better at bringing these points out.

*I wonder if the manuscript describing the development and testing of the tool (which was submitted together with this manuscript) could be combined into one manuscript*.

We had felt similarly and previously prepared a manuscript combining the tool and the findings presented in this current manuscript. Unfortunately, 18 months ago such a “merged” manuscript did not review well as frustrated both informaticists and biologists each desiring more detail. Thus we chose to separately publish the tool VirSorter: mining viral signal from microbial genomic data, S Roux, F Enault, BL Hurwitz, MB Sullivan, PeerJ 3, e985 – and here present its first application to ∼15K publicly available bacterial and archaeal genomes (this study).

[Editors’ note: what now follows is the decision letter after the authors submitted for further consideration.]

*1) Data availability: given that this is a Tools and Resources article, we feel that the data availability section should be more prominent. We suggest to move the availability section up (possibly before the conclusions) and provide a little bit more detail. iVirus.us itself seems like a rather hollow shell – clicking on data access yields a 502 bad gateway error. As far as I can tell, everything happens within the discovery environment of iPlant for which registration is necessary. Please elaborate a little bit. The MetaVir environment seems useful, but some of the MetaVir analysis haven't completed yet. In addition, we think that the* “*richly annotated genbank files*” *(promised in the rebuttal letter) should be made available not only on the author's website but uploaded to a big-data repository such as data dryad*.

We agree with the idea of placing more emphasis on data availability and appreciate the suggestions for how to do so. To this end, we have:

A) Created a “Dataset availability” section. This section is located at the end of the manuscript just before the conclusions, and now details the different places where the VirSorter Curated Dataset and the associated results are available.

B) Created a direct iVirus link for the datasets. As noted by the reviewers, the structure of iVirus is very young and still in development for the most part. However, we will leverage here a new feature in iVirus which allows for direct access to a set of files linked to a publication without the need for registration. This link provides direct access: http://mirrors.iplantcollaborative.org/browse/iplant/home/shared/ivirus/VirSorter_curated_dataset. We added this link to the manuscript in this new section (“Dataset availability”).

C) Made the annotated genbank files available via DataDryad. These are organized by host and provided as a zip package now uploaded to DataDryad (DataDryad package dryad.b8226) and added this information in the subsection “Dataset availability”of the revised manuscript.

*2) It seems the authors have misunderstood the request for making the scripts available. We were not asking for the VirSorter scripts, but the scripts that analyze the VirSorter data set to produce figures and results of the paper. Those scripts provide the most accurate description of the methods, and in the interest of reproducibility, they should – whenever possible – be made available. The preferred place would be a separate GitHub repository*.

Indeed, we misunderstood the former request from the reviewers. To rectify this, we have now prepared the scripts used to produce the results in this manuscript for public release on our lab wiki (the corresponding link: http://tmpl.arizona.edu/dokuwiki/doku.php?id=bioinformatics:scripts:vsb) and a GitHub repository (https://github.com/simroux/virsorter-curated-dataset-scripts-package).

*3) Host association figure: We continue to be underwhelmed by this figure. There are lots of lines which clearly fall into a handful of modules, but within these modules it is pretty hard to see what is going on. Maybe a two-way clustering would be more insightful. Consider a distance matrix d_ij, where d_ij is the fraction of sequences in viral cluster (VC) i that come from genomes of host phylum j, maybe normalized for the abundance of genomes from phylum j. Then cluster this matrix both by VC and host, similar to RNA-seq being clustered by gene and tissue. The modules should show up as blocks on the diagonal, while promiscuous affiliations are off-diagonal terms. What exactly the distance matrix d_ij should be requires some thought and there are probably better choices then this proposal. But if something like this would work out, it could be more informative than the current figure. Keeping one as a supplement of the other could be a good solution*.

We thank you for helping us see the issues with this figure better. To clarify these results, we modified the figures and represent the same network (and the same modules, identified through lp-brim) in a matrix form as suggested by the reviewers. Although we find that the overall “shape” of the network is not as apparent as in the “network” visualization, the “matrix” plot makes it indeed easier to identify the connections between virus clusters and host groups. The new figure (“matrix” visualization) was thus added as Figure 5, and the former Figure 5 is now displayed as Figure 5—figure supplement 1. We hope that these two representations together help present the findings in a manner that is most informative.